# Targeting of the NOX1/ADAM17 Enzymatic Complex Regulates Soluble MCAM-Dependent Pro-Tumorigenic Activity in Colorectal Cancer

**DOI:** 10.3390/biomedicines11123185

**Published:** 2023-11-30

**Authors:** Jimmy Stalin, Oriana Coquoz, Rachel Jeitziner Marcone, Stephane Jemelin, Nina Desboeufs, Mauro Delorenzi, Marcel Blot-Chabaud, Beat A. Imhof, Curzio Ruegg

**Affiliations:** 1Department of Pathology and Immunology, University of Geneva Medical School, Rue Michel Servet 1, CH-1211 Geneva, Switzerland; stephane.jemelin@unige.ch (S.J.); beat.imhof@unige.ch (B.A.I.); 2Department of Oncology, Microbiology, and Immunology, Faculty of Science and Medicine, University of Fribourg, Chemin du Musée 18, PER17, CH-1700 Fribourg, Switzerland; oriana.coquoz@unifr.ch (O.C.); nina.desboeufs@unifr.ch (N.D.); curzio.ruegg@unifr.ch (C.R.); 3C2VN, Inserm 1263, Inra 1260, UFR Pharmacie, Aix-Marseille University, 27 Bd J. Moulin, 13005 Marseille, France; marcel.blot-chabaud@laposte.net; 4Bioinformatics Core Facility, SIB Swiss Institute of Bioinformatics, CH-1015 Lausanne, Switzerland; rachel.marcone@sib.swiss (R.J.M.); mauro.delorenzi@unil.ch (M.D.)

**Keywords:** NADPH oxidases, melanoma cell adhesion molecule, angiogenesis, lymphangiogenesis, pharmacologic inhibitors, colorectal tumor, bioinformatic analysis

## Abstract

The melanoma cell adhesion molecule, shed from endothelial and cancer cells, is a soluble growth factor that induces tumor angiogenesis and growth. However, the molecular mechanism accounting for its generation in a tumor context is still unclear. To investigate this mechanism, we performed in vitro experiments with endothelial/cancer cells, gene expression analyses on datasets from human colorectal tumor samples, and applied pharmacological methods in vitro/in vivo with mouse and human colorectal cancer cells. We found that soluble MCAM generation is governed by ADAM17 proteolytic activity and NOX1-regulating ADAM17 expression. The treatment of colorectal tumor-bearing mice with pharmacologic NOX1 inhibitors or tumor growth in NOX1-deficient mice reduced the blood concentration of soluble MCAM and abrogated the anti-tumor effects of anti-soluble MCAM antibodies while ADAM17 pharmacologic inhibitors reduced tumor growth and angiogenesis in vivo. Especially, the expression of MCAM, NOX1, and ADAM17 was more prominent in the angiogenic, colorectal cancer-consensus molecular subtype 4 where high MCAM expression correlated with angiogenic and lymphangiogenic markers. Finally, we demonstrated that soluble MCAM also acts as a lymphangiogenic factor in vitro. These results identify a role for NOX1/ADAM17 in soluble MCAM generation, with potential clinical therapeutic relevance to the aggressive, angiogenic CMS4 colorectal cancer subtype.

## 1. Introduction

Among the hallmarks of cancer, tumor angiogenesis, the process of new blood vessel formation is crucial for solid-tumor growth, progression, and metastatic dissemination [1,2,3]. The absence of angiogenesis leads to dormant tumors, which can persist for years as microscopic dormant lesions [4,5,6]. Cancer cells, as well as cells of the tumor microenvironment (TME), are involved in the process of tumor angiogenesis. Tumor angiogenesis is promoted by the angiogenic switch, characterized by an imbalance between the production of pro-angiogenic versus anti-angiogenic factors, leading to the activation of quiescent vessels [7,8,9]. Among the TME cells, bone-marrow-derived monocytes (BMDM) and tumor-associated macrophages (TAM) promote this event. A plethora of soluble angiogenic factors (e.g., vascular endothelial growth factors (VEGFs), fibroblast growth factors (FGFs)), and their cognate receptors on endothelial cells (e.g., VEGF-Rs and FGF-Rs) have been identified [10]. Some of them have emerged as therapeutic targets and their inhibition by small molecules or blocking antibodies has led to the development of several drugs tested in pre-clinical studies or approved for clinical use (e.g., bevacizumab, sorafenib) [11,12]. Despite pre-clinical results demonstrating the eminent role of tumor-angiogenesis in promoting tumor growth, and successful tumor control with anti-angiogenic therapies, the clinical benefits have been rather limited and mostly seen only in combination with chemotherapy [13,14,15,16]. The reasons for this discrepancy are likely multiple, including intrinsic and acquired resistance to anti-angiogenic drugs, vascular cooption, and tumor adaptation [17,18,19]. Closing this gap between preclinical and clinical observation calls for a better understanding of tumor vascular biology, the mechanisms of tumor adaptation, and the identification of additional therapeutic target candidates with distinct mechanisms of action.

It has been reported that NADPH-oxidase family members, in particular NOX1 and NOX4, are expressed by cancer cells and promote tumor growth and metastasis in several cancers (i.e., melanoma, gastric, pancreatic, and colorectal cancers) [20]. They are also expressed by the cellular components of blood vessels (e.g., endothelial cells, pericytes, and vascular smooth muscle cells) and promote tumor angiogenesis [21,22,23,24]. Interestingly, the pharmacologic inhibition of NOX1, using a selective NOX1 inhibitor (GKT771) and NOX1 genetic ablation in mice, demonstrated potent anti-tumor and anti-angiogenic activities [25]. However, the molecular mechanisms by which NOX1 targeting affects tumor growth, angiogenesis, and the TME immune and inflammatory cell components are largely unknown. Recently, a direct physical association between NOX1 and ADAM17 was observed and ADAM17 degradation with a decrease in downstream signaling pathways via protein ubiquitination was reported after NOX1 silencing using siRNA [26]. ADAM17 is a well-characterized protease involved in a plethora of physiological and pathological processes including metabolic syndrome, diabetes, and cancer [27,28,29,30]. These reports extend previous observations involving ADAM17 in the generation of pro-angiogenic and pro-lymphangiogenic factors. Importantly, genetic deletion and pharmacological inhibition of ADAM17 reduce tumorigenesis and angiogenesis [31,32,33].

The soluble form of the melanoma cell adhesion molecule (sMCAM, sMUC18, or sCD146) has been reported as a novel promoter of tumor angiogenesis and tumor growth in experimental melanoma, ovarian and pancreatic cancers [34]. The therapeutic targeting of sMCAM using the monoclonal antibody M2J-1 decreased growth, angiogenesis, and the metastatic dissemination of human tumors in mice. These reports demonstrated that sMCAM promotes the metastatic dissemination of MCAM-positive cancer cells by inducing the epithelial-mesenchymal transition (EMT), and the expression of cancer stem cell markers and coagulation factors [35]. Importantly, the sMCAM is generated by proteolytic shedding of membrane MCAM (mMCAM) from the surface of both cancer cells and blood endothelial cells [36,37]. In endothelial cells, two metalloproteinases involved in this proteolytic cleavage were identified, ADAM17 and ADAM10. Both generate sMCAM from the short or the long mMCAM isoform, respectively. This specificity is mainly due to the concomitant cellular localization of ADAM17 with the short mMCAM isoform at the apical side of the endothelial cell surface, and ADAM10 with the long mMCAM isoform at the basolateral side of the cell surface [38]. However, the mechanism regulating mMCAM cleavage supported by ADAM17 under physiological and pathological conditions remains elusive at this point. Unraveling this mechanism is of potential clinical interest, as canonical protease inhibitors, including ADAM inhibitors, lack target specificity resulting in significant unwanted effects in clinical studies [28].

In this work, we decided to determine the involvement of NOX1 and ADAM17 in the generation of sMCAM from blood vascular endothelial and cancer cells and to demonstrate its contribution to tumor angiogenesis and cancer progression. We also investigated the potential clinical-pathological relevance of NOX1, ADAM17, and MCAM expression in colorectal cancer.

## 2. Materials and Methods

### 2.1. Endothelial Cell Isolation and Culture

Umbilical cords were collected for endothelial cell isolation. HUVECs were isolated following this procedure. Briefly, the cord vein was cleared by perfusion with PBS, followed by incubation with collagenase (1 mg/mL; Invitrogen, Carlsbad, CA, USA) in PBS for 15 min at 37 °C. The vein was then perfused with PBS to remove the endothelial cells and the suspension was centrifuged at 200× *g* for 5 min. The cell pellet was resuspended and maintained in complete M199 containing 10% fetal bovine serum (FBS), 15 µg/mL endothelial cell growth supplement (Upstate Biotechnology, Charlottesville, VA, USA), 100 µg/mL heparin (Sigma-Aldrich, St. Louis, MO, USA), 50 µM hydrocortisone (Sigma-Aldrich), and 10 µg/mL vitamin C (Sigma-Aldrich). Cells were cultured and used up to passage 5. Human dermal lymphatic endothelial cells (HDLECs) were from PromoCell and cultured in the provided cell culture medium.

### 2.2. Tumor and Immortalized Cell Line Cultures

DLD1 and Lovo human colorectal cancer cells and MC38 mouse colorectal cancer cells were cultured in DMEM (Life Technologies, Carlsbad, CA, USA) supplemented with 10% FBS, 1% penicillin-streptomycin solution, 1% L-glutamine, and 1% sodium pyruvate. Cells were grown at 37 °C in a humidified atmosphere containing 5% CO_2_. DLD1 and MC38 colorectal cancer cell lines were obtained from the Beat Imhof laboratory (Geneva University, Switzerland), and the Lovo colorectal cancer cell line was a gift from Dr. Franck Peiretti (Aix-Marseille University, France).

### 2.3. Immunoprecipitation of Proteins from Cell Lysates

Colorectal, melanoma cancer cell lines, or HUVEC were lysed in cooled NP-40 lysis buffer containing protease inhibitor cocktail (PI) and phosphatase inhibitors (phenylmethyl-sulfonyl fluoride, sodium orthovanadate, and β-Glycerophosphate disodium salt hydrate). Protein lysates were kept under agitation for 10 min at 4 °C and centrifuged at 13,000 rpm for 10 min at 4 °C to remove cellular debris. Protein lysates were incubated with the corresponding primary antibodies under constant rotation overnight at 4 °C. Afterward, protein lysates were incubated under constant rotation with prewashed magnetic beads of protein G (Sino Biological, Wayne, PA, USA) for 2 h at 4 °C. Then, magnetic beads bound to the protein complex of interest were washed three times with cell lysis buffer containing PI. Magnetics beads were eluted in 50 μL of cell lysis buffer and analyzed by Western blot.

### 2.4. Cells Pharmacological Drug Treatments

Cells were treated with TMI-5 (Apratastat, Irvine, CA, USA), TMI-1, TAPI-1, GKT771, or GKT831 (Setanaxib) at a dose of 10 µM for 4 or 24 h. The supernatant was collected and centrifuged at 12,000 rpm for 10 min to remove any cellular debris while adherent cells were rinsed gently with PBS once followed by lysis for activity assays, protein, or RNA analysis.

### 2.5. Western Blot

After protein quantification with the Pierce BCA protein assay kit (ThermoScientific, Waltham, MA, USA), lysates were boiled with the LDS sample-reducing loading buffer (Thermofisher Scientific) and subjected to SDS-polyacrylamide gel electrophoresis. Proteins were electroblotted onto a PVDF membrane. Membranes were exposed to specific antibodies overnight under agitation at 4 °C. Then after three wash steps, blots were exposed to specific secondary antibodies for 1 h under agitation at 4 °C and revealed with a horseradish peroxidase-conjugated secondary antibody (Jackson Laboratories Inc., West Grove, PA, USA) or with fluorescent Dylight 680 and 800 secondary antibodies and finally read with a fluorescence blotting reader (Licor, Lincoln, NE, USA, Odyssey CLx).

### 2.6. ADAM17 Activity Measurement

ADAM17 activity in cell lysates was determined using the Sensolyte 520 TACE activity assay kit fluorometric (AnaSpec, Fremont, CA, USA) following the manufacturer’s instructions).

### 2.7. Soluble MCAM ELISA

Platelet-diminished plasma was prepared from mouse blood using a two-step centrifugation method or cell culture supernatant of HUVEC and cancer cells were assayed using the corresponding CY-QUANT soluble CD146 ELISA kit (Biocytex, Marseille, France) and the mouse soluble MCAM ELISA kit (Fisher Scientific) following manufacturer’s instructions.

### 2.8. Quantitative Reverse Transcription-Polymerase Chain Reaction (qRT-PCR)

Total RNA from tumor samples was extracted using the Nucleospin RNA Extraction Kit (Macherey-Nagel, Oensingen, Switzerland). Extracted total RNA was quantified by measuring the OD at 260 nm using a NanoDrop spectrophotometer (Thermo Fisher Scientific). Reverse transcription (RT) was then carried out with 1 μg total RNA in a reaction mixture containing RT buffer, dNTPs, random primer, MultiScribe reverse transcriptase, and H_2_O to a final volume of 10 µL, according to the manufacturer’s protocol (Applied Biosystems, CA, USA). The RT reaction was performed in a thermocycler as follows: 25 °C for 10 min, 37 °C for 2 h, and 85 °C for 5 min followed by a hold at 4 °C. For RT-PCR, 50 ng/μL cDNA was added to 24 μL of a mixture consisting of 12 µL SYBR Green (Agilent/Stratagene, Santa Clara, CA, USA), forward and reverse primers and H_2_O. PCR was then carried out on a Stratagene Mx 3000 Pro QPCR device (Agilent Technologies) as follows: one cycle for 10 min at 95 °C, followed by 40 cycles for 30 s at 95 °C and 1 min at 60 °C. The results were analyzed using the MxPro software version 4.10 (Stratagene). qRT-PCR data are presented as ∆Ct values of the genes of interest relative to the housekeeping gene GAPDH.

### 2.9. Animal Procedures

C57BL/6J immunocompetent mice and NSG immunodeficient mice were purchased from Janvier Labs (Le Genest Saint Isle, France) and the Jackson Laboratory, respectively. NOX1-deficient mice (B6.129X1-Nox1tm1Kkr/J) were purchased from the Jackson Laboratory. All animal procedures were performed according to the guidelines set down by the Institutional Ethical Committee of Animal Care and the Swiss Cantonal Veterinary Office of Geneva and Fribourg (authorization number GE93/14 and GE150/17 for Geneva and 2016_06_FR for Fribourg). Male mice aged between 5 and 7 weeks were used.

### 2.10. Analysis of Tumor Growth in Xenograft-Bearing Animals

Mouse and human tumor cell line xenografts were produced by subcutaneous injection of tumor cell suspensions in PBS (5 × 10^5^ cells for DLD1, Lovo, and MC38) into the back of mice. When the tumors reached 50 mm^3^, mice received a twice-weekly intraperitoneal injection of mAbs at a dose of 100 µg per mouse (anti-soluble MCAM, bevacizumab, and control IgG antibodies), or Apratastat (ADAM17 inhibitor) at 10 mg/kg daily by oral gavage in a methylcellulose/Tween 80 solution (vehicle) until sacrifice.

Tumor size was measured twice per week using a caliper, and tumor volume was determined according to the equation: length × width × thickness × 0.5236. Tumor size and mass were also evaluated after sacrifice. At the time of sacrifice, animals were anesthetized with a mix of ketamine and xylazine (80 mg/kg and 10 mg/kg, respectively) before intra-tracheal instillation of 10% formalin to fix the tissues for future analysis.

### 2.11. Flow Cytometry

After dissection, tumors were diced with razor blades and digested using a gentleMACS Dissociator and reagents from the corresponding mouse and human tumor dissociation kits (Miltenyi Biotech, Teterow, Germany). Single tumor cell suspensions were obtained by straining through a 70 µm mesh filter, after which the strained cells were washed twice in FACS buffer (PBS/Fetal Calf Serum 5%/5 mM EDTA). Cells were incubated for 30 min at 4 °C with an anti-CD16/CD32 Fc blocking antibody (BD Biosciences, San Jose, CA, USA) to block Fc receptor binding and then washed once with FACS buffer. The cells were stained with the indicated fluorophore-conjugated antibodies and analyzed on a Gallios flow cytometer (Beckman Coulter, Brea, CA, USA) to quantify the proportion of blood vascular endothelial cells and lymphatic vascular endothelial cells, macrophages and tumor-associated macrophages, and lymphocyte populations in the tumors, as previously described [25].

### 2.12. Mouse Aortic Ring Assay

Thoracic aorta was dissected and adipose tissue surrounding the aorta was removed from 6–8-week-old C57BL/6 mice and NOX1 KO mice [39]. Then, the aortas were cut with a scalpel into 0.5 mm wide rings. The rings were then incubated overnight at 37 °C in 5% CO_2_ in serum-free OptiMEM (Life Technologies). The following day, the rings were embedded in type I collagen (1 mg/mL, Millipore, Burlington, MA, USA) and maintained at 37 °C in a 5% CO_2_, 95% humidity incubator in growing medium (i.e., OptiMEM supplemented with 2.5% FCS, 100 U/mL penicillin/streptomycin, and 30 ng/mL VEGF165 (Peprotech, Cranbury, NJ, USA)). Then, aortic rings were cultured in the presence of the different ADAM17 and metalloprotease inhibitors (TMI-1, TMI-5, and TAPI-1). Daily microvessel outgrowth was imaged by phase-contrast microscopy (ImageXpress; Molecular Devices, San Jose, CA, USA) and subsequently quantified with MetaMorph (Molecular Devices). The nonsprouting fragments were excluded from the quantification.

### 2.13. Bioinformatic Analysis of Colorectal Tumor Datasets

The Kaplan–Meier method in the R library ‘survminer’ (Drawing Survival Curves software, ‘ggplot2’ R package version 0.4.4.) was used to generate relapse-free and overall survival curves. For our purpose, patients were stratified according to the expression of MCAM, NOX1, and ADAM17 relative to the median value. Expression data and clinical data were extracted from the Cancer Genome Atlas Research Network (https://www.cancer.gov/tcga, accessed on 15 June 2021), PETACC3, and the GEO repository entry GSE39582. *T*-tests with equal variance were used to compare the mRNA expression in different subgroups of patients.

Pearson’s correlation and Spearman’s rank correlation coefficients were calculated to analyze the relationships between MCAM expression and genes of interest. Different tested gene sets were used including angiogenesis genes (gene ontology, GO:0001525), genes known to be positively (GO:0045766) or negatively (GO:0016525) associated with angiogenesis, genes associated with angiogenesis according to published data, and gene sets known to be positively associated with tumor-associated macrophage signatures (GO:0006955). In addition, tested gene sets included immune-related genes (GO:0006955), anti-tumor immune response-related genes (GO:0002418), genes related to immune system processes (GO:0002376), and inflammatory genes associated with activated macrophages. The association between MCAM expression and other genes was quantified by calculating the odds ratio (OR) using Fisher’s test, with expression stratified based on expression relative to the median value. Confidence intervals were obtained under the assumption that the log (OR) followed a normal distribution.

The R library ‘ggplot2’ (https://www.springer.com/gp/book/9780387981413, accessed 25 June 2021) was used to generate forest and correlation plots. All statistical analyses were performed using R software (version 3.5.3).

### 2.14. NanoString nCounter mRNA Assay

The amount of mRNA content in tumor biopsies was determined using the nCounter Sprint Profiler platform (NanoString Technologies, Seattle, WA, USA) and analyzed with nSolver analysis Software version 4.0 on the iGE3 Genomics Platform of the University of Geneva.

### 2.15. Cell Proliferation Assay

HDLECs were seeded on 96-well plates (5 × 10^3^ cells/well). They were cultured in EGM-2-mv medium for 1 day. Cells were then preincubated for 3 h in EBM-2 without growth factor medium. Finally, cell proliferation was assayed by quantification of the incorporation of 5-bromo-2′-deoxy-uridine (BrdU) into cellular DNA using the BrdU labeling and detection kit III (Roche Applied Science, Basel, Switzerland). In brief, cells were incubated for 12 h with BrdU-labeling solution in EBM-2 medium in the absence or presence of rsCD146. Then, the cells were fixed and cellular DNA was partially digested by nuclease treatment and incorporated BrdU was detected with a peroxidase conjugated anti-BrdU primary antibody. The absorbance was measured at 450 nm using a microplate reader (Tecan, Männedorf, Switzerland). Results were expressed as arbitrary units. Experiments were performed in triplicates.

### 2.16. Cell Death Detection

The extent of histone-associated DNA fragments in the cytoplasmic fraction of cell lysate was measured by Cell Death Detection ELISA plus Kit (Roche, Meyla, France) as described by the manufacturer.

### 2.17. Wound Healing

To assay migration, we performed the wound-healing assay. Briefly, HDLECs were plated in Matrigel-coated 96-well plates. One day after plating, the cell monolayer was scratched to make a regular wound. Cells were allowed to migrate for 12 h. The migration area was then measured and calculated using the MetaXpress software (https://www.moleculardevices.com/products/cellular-imaging-systems/acquisition-and-analysis-software/metaxpress (accessed on 1 June 2020), Molecular Device, Sunnyvale, CA, USA).

### 2.18. Endothelial Cell Tube Formation in Matrigel

The 96-well plates were precoated with a 1:1 mixture of cold Matrigel Basement Membrane (10 mg/mL, BD Biosciences): EBM-2 medium. After 45 min of polymerization at 37 °C, HDLECs were plated at 104 cells/well in EBM-2 supplemented or not with recombinant soluble MCAM or recombinant VEGF. After 6 h, pictures of representative fields were taken for each condition under an inverted microscope at 400× magnification. Capillary tube formation was evaluated by measuring the total number of tubes per field with the ImageJ software (https://imagej.net/ij/ij/) accessed in July 2021.

### 2.19. Statistical Analysis and Expression of Results

Data are expressed as the mean ± SEM of the indicated number of experiments. Statistical analysis was performed using Prism software version 8 (GraphPad Software, La Jolla, CA, USA). Before statistical analysis, the variance between groups was estimated. A non-parametric one-way ANOVA, followed by a Dunn’s multiple comparison test, was used when comparing more than two groups. Significant differences between the two groups were determined using the unpaired Student’s *t*-test or the Mann–Whitney test. A *p* value ≤ 0.05 was considered significant. For the animal studies, the investigator was blinded to group allocation, and all mice were distributed randomly among the various groups.

## 3. Results

### 3.1. NOX1, ADAM17, and Membrane MCAM Interact in a Molecular Complex

It has been previously reported that NOX1 interacts with ADAM17 and protects it from ubiquitin-mediated degradation in human colon cancer cells [26]. In parallel, ADAM17 interacts with short mMCAM, inducing its proteolytic cleavage and the release of sMCAM from the endothelial cell surface into the cell culture medium [38]. Independently, we previously observed that either NOX1 or sMCAM pharmacological inhibition impaired tumor growth and angiogenesis [25]. These observations suggest that in endothelial and cancer cells NOX1, ADAM17, and mMCAM may associate in a tri-molecular functional complex contributing to the enzymatic function of ADAM17 and sMCAM release. To test these hypotheses, we monitored NOX1, ADAM17, and mMCAM protein expression and performed co-immunoprecipitation (co-IP) experiments from whole lysates of human endothelial cells (HUVEC) and human colorectal cancer (CRC) cell lines (HCT-116, DLD1, and SW480). We observed that HUVEC expressed a higher level of mMCAM protein compared to the three CRC cell lines, while CRC cells expressed slightly higher levels of ADAM17 and NOX1 proteins relative to HUVEC (Figure 1A). The IP of mMCAM, NOX1, and NOXO1 from HUVEC and DLD1 cells, followed by Western blotting, revealed the presence of co-immunoprecipitated ADAM17, NOX1, and mMCAM proteins in all tested cell lines (Figure 1B). Similar results were obtained using the MC38 CRC and B16F10 mouse melanoma cell lines (Figure 1C). Conversely, immunoprecipitation of ADAM17 or short mMCAM isoform also co-immunoprecipitated NOX1 and mMCAM proteins in HUVEC and MC38 cell lines, as revealed by subsequent Western blotting (Figure 1D).

From these experiments, we conclude that NOX1, ADAM17, and mMCAM are present together in a complex in endothelial and CRC cell lines. We especially show that the short membrane MCAM isoform is present in this complex.

### 3.2. NOX1 and ADAM17 Inhibition Decreases Soluble MCAM Release from Endothelial and Colorectal Cancer Cells

As the protease ADAM17 is involved in the generation of sMCAM, we decided to study how pharmacologic inhibition of ADAM17 could affect sMCAM shedding. The ADAM17 inhibitors TMI-1 and TMI-5 decreased the ADAM17 activity in HUVEC lysates and the sMCAM concentration in HUVEC supernatants after 4 h and 24 h (i.e., short- and long-term) treatments. This validated the idea that the proteolytic activity of ADAM17 is necessary for the shedding of mMCAM as soluble sMCAM (Figure 2A,B).

As NOX1 regulates ADAM17 expression and activity by preventing its degradation [26], we were interested in determining if NOX1 pharmacological inhibition may decrease ADAM17 activity and sMCAM released from endothelial cells. Therefore, we measured ADAM17 activity in cell lysates and sMCAM concentration in the supernatants of HUVECs treated for 4 or 24 h with GKT771, a specific NOX1 inhibitor, and GKT831, a NOX1/NOX4 inhibitor. We observed a decrease in cell-bound ADAM17 activity after 24 h of treatment while the 4 h treatment was ineffective (Figure 2C). The measurement of sMCAM protein concentration in cell culture supernatant consistently revealed a decrease in sMCAM concentration in HUVEC culture supernatant after 24 h treatment while 4 h treatments did not affect sMCAM release (Figure 2D). TAPI-1, a pan-metalloprotease inhibitor, used as a positive control led to a strong but partial inhibition of ADAM17 activity and sMCAM release after both short- and long-term treatments. To collect evidence for the in vivo relevance of these results, we measured sMCAM levels in the plasma of WT C57BL6/J mice bearing MC38 tumors treated with GKT771, GKT831, or vehicle compound. In parallel, MC38 tumor-bearing NOX1-deficient C57BL6/J mice were also treated with GKT771 or vehicle compound. Interestingly, we found that GKT771 and GKT831 inhibitors and NOX1 genetic deletion partially decreased sMCAM plasma concentration compared to vehicle-treated or WT mice, respectively (Figure 2E).

From these experiments, we concluded that long-term (i.e., 24 h) pharmacological inhibition of NOX1 and NOX4 partially decreases ADAM17 activity and sMCAM release from endothelial cells while both short- and long-term ADAM17 inhibition partially decrease sMCAM release in vitro. Consistently, pharmacological NOX1 inhibition, or NOX1 gene deletion, results in a decreased mMCAM plasma level in tumor-bearing mice.

### 3.3. NADPH Oxidase Pharmacological Inhibitors Impair ADAM17 and MCAM Expression in Endothelial Cells

As the 4 h treatment with NOX pharmacological inhibitors did not affect ADAM17 activity and sMCAM release while the 24 h treatment did, we aimed to determine if long-term pharmacological treatment would have an influence on ADAM17 and mMCAM expression on HUVECs. To this end, we performed quantitative PCR experiments on HUVECs treated with the NOX pharmacological inhibitors. We found that NOX pharmacological treatments of endothelial cells decreased ADAM17 and mMCAM mRNA expression (Figure 2F).

These results demonstrate that NOX pharmacological inhibition decreases ADAM17 and membrane MCAM mRNA expression in endothelial cells, and this will also contribute to the decrease in soluble MCAM release.

### 3.4. ADAM17 and sMCAM Targeting Inhibits Mouse CRC Tumorigenic and Angiogenic Properties Downstream of NOX1

As NOX1 is expressed by tumor cells and host cells, in particular endothelial cells, and regulates ADAM17 activation and sMCAM release, we next assessed whether ADAM17 or sMCAM tumor-promoting functions would act downstream of NOX1.

To address this question, we performed mouse MC38 (ADAM17-positive CRC cells) tumor growth-inhibition studies with TMI-5 (Apratastat) in both WT and NOX1-deficient mice. In vivo tumor growth and ex vivo tumor size analyses revealed that treatment with TMI-5 prevents MC38 tumor growth in WT mice as compared to IgG or vehicle-treated mice (Figure 3A), as well as in NOX1-deficient mice as compared to WT mice. Pharmacological ADAM17 inhibition with TMI-5 resulted in additive inhibitory effects on tumor growth despite NOX1 deficiency in the mice (Figure 3A).

As ADAM17-mediated promotion of angiogenesis was reported to involve sMCAM release [31,33,38,40], we aimed to unravel how ADAM17 inhibition could also affect MC38 mouse colorectal tumor angiogenesis and lymphangiogenesis. The flow cytometry analysis of tumor-derived single-cell suspensions revealed a decreased percentage of CD31+/GP38- blood vascular endothelial cells (BEC) and CD31+/GP38+ lymphatic endothelial cells (LEC) in WT tumor-bearing mice treated with TMI-5, in NOX1-deficient mice and NOX1-deficient mice treated with TMI-5, as compared to vehicle-treated-WT mice (Figure 3B). TMI-5 treatment had no additive effect in tumor-bearing NOX1-deficient mice as compared to the corresponding vehicle-treated mice (Figure 3B). To confirm whether TMI-5-based ADAM17 inhibition may have a direct in vivo anti-angiogenic effect, independent of the presence of a tumor, we tested the ability of TMI-1 and TMI-5 to inhibit endothelial cell sprouting in the ex vivo aortic ring model. We observed a significant decrease in the branching area in aortic rings treated with both TMI-1 and TMI-5 inhibitors as well as in TAPI-1-treated aortic rings relative to DMSO-treated aortic rings (Figure 3C).

Previously, we reported that the pharmacological treatment of immunodeficient mice bearing relatively rare human pancreatic and melanoma tumors with blocking anti-sMCAM monoclonal antibody (mAb) M2J-1 resulted in the efficient inhibition of tumor angiogenesis and tumor growth [34,35]. Whether the anti-tumor properties of anti-sMCAM mAb may also extend to colorectal carcinoma (CRC) remains unknown. To test this, we performed mAb-based blocking of sMCAM in C57BL6/J WT and NOX1-deficient mice bearing mouse MC38 CRC tumors. Treatment with blocking anti-sMCAM mAb M2J-1 reduced MC38 tumor growth and tumor size relative to control IgG (Figure 3D). However, anti-sMCAM mAb treatment did not further reduce tumor growth in NOX1-deficient mice as compared to WT mice (Figure 3D).

These results suggest that ADAM17 inhibition by TMI-5 has potent anti-angiogenic and anti-lymphangiogenic properties that are downstream of host NOX1. Furthermore, TMI-5 displays potential additive anti-tumor properties independent of host NOX1. Finally, sMCAM targeting blocks NOX1-dependent CRC tumor growth. These observations confirm that NOX1 is an upstream regulator of ADAM17 activity and sMCAM release in vivo.

### 3.5. Inhibition of Tumor ADAM17 Suppresses Tumor Growth and TAM Recruitment

In the above experiments (Figure 3A), we observed that ADAM17 inhibition further suppressed tumor growth in NOX1-deficient mice, suggesting a direct effect on cancer cells. To address this hypothesis and determine if ADAM17 inhibition is dependent on ADAM17 expression by cancer cells, we generated tumors with human DLD1 CRC cells expressing functional mature ADAM17 and human LoVo CRC cells expressing immature, inactive ADAM17 due to a deficiency in the proprotein convertase furin required for the processing of pro-ADAM17 into active ADAM17 [34,35]. Tumor-bearing mice were then treated with the ADAM17 inhibitor TMI-5. These experiments revealed that ADAM17 pharmacologic inhibition reduced DLD1, but not LoVo tumor growth, as compared to control treatments (Figure 3E,F). We previously demonstrated that NOX1 pharmacologic inhibition did not affect tumor angiogenesis in immunodeficient mice [25]. Flow cytometry analysis of vascular CD31+/GP38- and lymphatic CD31+/GP38+ tumor-associated endothelial cells in mice treated with TMI-5 led to similar results (Figure 3G,H). The analysis of inflammatory and immune cell populations revealed that the percentage of CD3+/CD4+, CD3+/CD8+, CD3+/NK1.1+ T lymphocytes, natural killer T lymphocyte subsets, and CD3-/B220+ B lymphocytes were unchanged in TMI-5-treated DLD1 tumors (Appendix A) while CD11b+/F4-80+ macrophages and CD11b+/F4-80+/CD68+ tumor-associated macrophages (TAM) were decreased (Figure 3I). The decrease in macrophages and TAM was also observed in the MC38 tumor model (Appendix A). No effects were observed in ADAM17 dysfunctional LoVo tumors (Figure 3I).

These results demonstrate that the anti-tumor effects of ADAM17 pharmacological inhibition also involve the targeting of the ADAM17 expressed by cancer cells in addition to the host-associated anti-angiogenic effect observed previously. Furthermore, ADAM17 pharmacologic inhibition also decreases TAM recruitment.

### 3.6. Targeting Soluble MCAM Impairs Human CRC Tumor Growth

We previously demonstrated that human cancer cells released sMCAM and that sMCAM inhibition suppressed tumor growth [34,35]. Therefore, we investigated the potential effect of NOX1 and ADAM17 inhibition on the generation of sMCAM by mouse melanoma and CRC cells. Consistent with the results in HUVEC, we observed a decreased sMCAM protein concentration in MC38 and B16F10 supernatants after 24 h of treatment with GKT771, GKT831, or TAPI-1 (Figure 4A).

To determine if the anti-tumor effect of anti-sMCAM mAb treatment was dependent on MCAM expression by the tumor cells themselves or by the host, we treated immunodeficient NSG mice bearing either human DLD1 (mMCAM-positive) or LoVo (mMCAM-negative) CRC tumors with anti-sMCAM mAb. The in vivo tumor growth analysis and ex vivo tumor size measurement revealed that anti-sMCAM mAb treatment decreased DLD1 and LoVo tumor growth as compared to IgG treatments similar to an anti-angiogenic drug (i.e., bevacizumab), which targets human VEGF-A (Figure 4B,C).

These results demonstrate that tumor cells release soluble MCAM downstream of NADPH oxidase activity. However, treatment with anti-sMCAM mAb decreases human CRC tumor growth in immunodeficient mice independently of mMCAM expression by tumor cells.

### 3.7. MCAM mRNA Expression Is Elevated in the Angiogenic CMS4 Human CRC Subtype

To collect clinical evidence for a relationship between NOX1, ADAM17, and MCAM, we characterized their expressions in human CRC. We analyzed the association between clinical-pathological characteristics and the expression of NOX1, ADAM17, and MCAM mRNA in three datasets (GSE39582, The Cancer Genome Atlas, and PETACC3). We observed a tendency for an association between the high expression of MCAM mRNA and both shorter relapse-free survival (RFS) and overall survival (OS) (Figure 5A). However, no trend or association was found for NOX1 and ADAM17 (Appendix A). Higher MCAM and ADAM17 mRNA expressions were detected in advanced-stage tumors (stages 3 and 4), as compared to early-stage tumors (stages 1 and 2). NOX1 mRNA expression was unchanged across the stages (Appendix A). Furthermore, we found that high tumor NOX1 mRNA expression was associated with WT KRAS while both low ADAM17 and MCAM mRNA expression was associated with mutated KRAS (Appendix A).

At the genomic level, CRC can be classified into chromosomal stable/microsatellite instable (MSI) hypermutated tumors and chromosomal instable (CIN)/microsatellite stable (MSS) tumors. At the transcriptomic level, CRC can also be classified into four different consensus molecular subtypes (CMS 1–4) [41]. Interestingly, stratification of MSI versus MSS tumors revealed elevated expression of NOX1, ADAM17, and MCAM mRNA in the MSS subtype as compared to the MSI subtype (Figure 5B). MCAM mRNA expression was higher in the CMS4 subtype as compared to the other three subtypes, while NOX1 mRNA was elevated in CMS 2–4 subtypes (Figure 5C).

These results demonstrate that elevated NOX1, ADAM17, and MCAM mRNA expression in CRC patients is dominant in the MSS tumor subtypes. Strikingly, elevated MCAM expression is found in the CMS4 angiogenic tumor subtypes and advanced-stage disease (i.e., stages 3 and 4).

### 3.8. MCAM Expression Correlates with VE-Cadherin and Pro-Angiogenic Factors in Human CRC

The CMS4 CRC subset is associated with angiogenesis and stromal cell invasion [41]. We found that NOX1, ADAM17, and MCAM mRNA expression were elevated in the CMS4 subgroups, and we assessed the correlation between MCAM mRNA and angiogenic factor expression in several CRC mRNA expression datasets (GO: 0001525, GO: 0045766, and GO: 0016525) [42]. We observed a positive correlation between MCAM mRNA expressions and the angiogenesis-associated factors CDH5, TIE1, TEK, KDR, PECAM1, ANGPT2, FLT1, ANGPT1, but not with VEGF-A (Figure 6A). Especially, a robust correlation was found between MCAM and CDH5/VE-cadherin mRNA expressions (Figure 6B). An extension of the analysis revealed that it also strongly correlates with pericyte markers including PDGF receptor β, PDGF-B, PDGF receptor α, and PDGF-A (Figure 6C).

### 3.9. MCAM Correlates with VEGF-C and FLT4 (VEGFR-3) Expression and sMCAM Promotes Lymphangiogenesis

Regarding the pro-angiogenic and lymphangiogenic roles of NOX1 and ADAM17 in CRC, we hypothesized that sMCAM could act as a lymphangiogenic growth factor and promote tumor lymphangiogenesis. To test this hypothesis, we first investigated if MCAM mRNA expression correlates with lymphangiogenic molecules. We observed that MCAM mRNA expression correlated with VEGFC and its receptor’s, FLT4/VEGFR-3, expression in CRC patients (Figure 7A). To experimentally support this observation, we monitored the effect of anti-sMCAM mAb on the expression of the selected mouse (host) and human (tumoral) transcripts in homogenates of DLD1 tumors growing in NSG mice by NanoString arrays (Appendix A). We observed a decrease in mRNA expression of vascular and angiogenic molecules such as mouse Pecam1, Mcam, Pdgf-β receptor, Vegf-a, and the lymphangiogenic growth factors Vegf-c in tumor-bearing mice treated with anti-sMCAM antibody relative to IgG (Figure 7B). These results demonstrate that MCAM expression correlates with pro-angiogenic and pericyte growth factors.

To functionally test whether sMCAM displays pro-lymphangiogenic properties, we monitored the effects of recombinant sMCAM (rs-MCAM) on human dermal lymphatic endothelial cells (HDLEC) in vitro (Appendix A). We observed that rh-sMCAM increased HDLEC proliferation in a dose-dependent manner (Figure 7C) and protected HDLEC against apoptosis (monitored by detecting DNA fragmentation) induced by H_2_O_2_ (Figure 7D). Finally, rh-sMCAM increased the in vitro lymphangiogenic ability of HDLEC by increasing their cellular motility in a wound-healing assay (Figure 7E) and the capacity to form a pseudo-capillary network using a 2-dimensional pseudo-capillary network assay (Figure 7F and Appendix A).

Taken together, these results demonstrate that MCAM expression correlates with lymphangiogenic markers in CRC tumors. Moreover, we reported that anti-sMCAM mAb treatment decreased Vegf-c mRNA expression in tumors while rh-sMCAM stimulated HDLEC proliferation, motility, and lymphangiogenic ability. Furthermore, sMCAM protects HDLEC against H_2_O_2_-induced apoptosis. These results are consistent with the pro-lymphangiogenic roles of ADAM17 and NOX1.

## 4. Discussion

In this study, we report that (i) MCAM, NOX1, and ADAM17 are associated in a protein complex in endothelial and cancer cells, and both ADAM17 and NOX1 regulate the shedding of membrane MCAM and the release of soluble MCAM, an angiogenic tumor growth factor; (ii) ADAM17 targeting reinforces the anti-tumor, anti-angiogenic, and anti-lymphangiogenic effects observed previously with NOX1 inhibition. It also decreases the recruitment of TAM, and this effect depends on ADAM17 expression by cancer cells; (iii) the targeting of soluble MCAM decreases CRC tumor development in immunocompetent and immunodeficient mouse models; (iv) MCAM expression is elevated in the angiogenic CMS4 subtype of CRC patients and correlates with angiogenic and lymphangiogenic growth factor expression.

The involvement of MCAM during tumor development and angiogenesis has already been studied [43,44]. However, the recent discovery of distinct intracellular signaling pathways of the two membrane MCAM isoforms (mMCAM) and the soluble form of MCAM (sMCAM) has renewed interest in the contribution of each form of MCAM to physiological and pathological processes [45,46,47]. The role of sMCAM during physiological or tumor angiogenesis, tumor growth, and metastatic dissemination has been especially highlighted recently [34,35].

Soluble MCAM is a pro-angiogenic factor, produced by endothelial and MCAM-positive cancer cells. Targeting sMCAM by blocking monoclonal antibodies decreases the growth of human melanoma, pancreatic, and ovarian carcinoma, as well as reducing metastatic dissemination [34,35]. Despite all the reports related to soluble MCAM functions in cancer, many questions remain unanswered, especially those concerning the functional relevance of sMCAM and the mechanism behind its shedding from cancer cells and vascular endothelium. Furthermore, the clinical and pathological relevance of MCAM expression in CRC is still unclear.

Here, we investigated the potential therapeutic properties of anti-sMCAM mAb on mouse MC38 cells, and two human CRC cancer cell lines (DLD1, membrane MCAM positive, and Lovo, membrane MCAM negative). Interestingly, sMCAM targeting reduces mouse and human CRC-tumor growth. This is the first demonstration of the anti-tumor effect of anti-sMCAM mAb on the development of colorectal tumors, especially in an immunocompetent experimental tumor model. This observation is of particular interest as it is closer to the human setting, suggesting that sMCAM is a potentially a clinically relevant therapeutic target. Of interest, anti-soluble MCAM mAb decreased the growth of human MCAM-negative Lovo tumors, suggesting that targeting sMCAM produced by the host is also therapeutically relevant. This result was corroborated by the analysis of mAb efficiency in NOX1-deficient mice, which demonstrated that the absence of host NOX1 impaired the anti-tumor effect of the anti-sMCAM antibody. From in vivo experiments, we observed that the blood concentration of sMCAM was decreased in NOX1-deficient mice and mice treated with NOX1-pharmacologic inhibitors, supporting the notion that NOX1 regulates sMCAM release.

These results are of clinical interest. Indeed, sMCAM inhibition has been described preferentially on membrane MCAM-positive tumors and the researchers suggested that anti-sMCAM therapeutic strategy could be extended to membrane MCAM-negative tumors by targeting host-derived sMCAM. These MCAM-positive blood cells could be constituted by vascular endothelial cells, pericytes, smooth muscle cells, T and B lymphocyte subsets, and/or mesenchymal stem cells [48,49,50,51].

Our findings demonstrate that ADAM17 constitutes a leading actor mediating the generation of sMCAM. Using in vivo models, we found that inhibition of ADAM17 recapitulated the main effects relative to NOX1 blockade but with increased efficiency. Indeed, inhibition of ADAM17 had a similar effect on MC38 and DLD1 tumors expressing functional ADAM17 in WT mice as compared to blocking sMCAM and NOX1. Its therapeutic effects persisted in NOX1-deficient mice but not in mice bearing Lovo tumors with non-functional ADAM17. These experiments indicate that ADAM17 inhibition directly affects MC38 and DLD1 tumor cells, but not LoVo cells lacking mature ADAM17 due to furin deficiency [52]. This direct effect on tumor cells also had the unique property of decreasing intra-tumor TAM recruitment in MC38 and DLD1 tumors in mice, while this effect was absent in the LoVo tumor model.

These observations are consistent with and extend experimental and pre-clinical investigations based on pharmacological targeting of sMCAM, NOX1, and ADAM17 or gene silencing approaches that have demonstrated growth inhibition of multiple types of tumors [25,34,35,53,54,55,56,57]. We found that targeting sMCAM recapitulates most of the anti-tumor effects mediated by NOX1 targeting and involves the effects of the immune tumor microenvironment (TME). Moreover, the absence of sMCAM-based therapeutic effects in NOX1-deficient mice suggests that it is a downstream regulator of NOX1 pro-tumorigenic and pro-angiogenic effects. Interestingly, we demonstrated that targeting ADAM17 encompasses the NOX1-sMCAM pathway but offers additional anti-tumor effects that regulate TAM recruitment.

The generation of sMCAM from membrane MCAM has been demonstrated by the presence of a shorter MCAM protein lacking TM and IC domains in the blood and in cell culture supernatants [36]. Its presence is associated with several pathologies including cancer and inflammatory disorders [35,44,58,59,60,61,62]. The generation of sMCAM depends on metalloproteases from the ADAMs family. Indeed, it has been recently demonstrated that ADAM17 and ADAM10 cleave the short and long mMCAM cell surface isoforms, respectively [38]. However, here, we newly report that the generation of sMCAM is impaired by inhibiting ADAMs activity or modulating the ADAM17 regulatory molecule NOX1 [26]. In addition, we demonstrate that NOX1, ADAM17, and membrane MCAM are present in a multiprotein complex in endothelial and cancer cells and that long-term treatment with NOX1 inhibitor partially decreases ADAM17 expression and the generation of sMCAM. These results corroborate previous reports showing that the silencing of NOX1 also decreases ADAM17 protein expression and allows proteolytic degradation of ADAM17 [26]. In turn, it reduces the shedding of MCAM, and the release of the intracellular domain involved in the de novo synthesis of membrane MCAM [38,45]. The decrease in ADAM17 activity and soluble MCAM release following NOX1 inhibition occurs only after prolonged exposure to the pharmacological inhibitors (e.g., 24 h), suggesting that the process takes place, and that inhibition of NOX1 activity, at least in a short period (e.g., 4 h), does not affect this mechanism. The process involved a decrease in ADAM17 and MCAM RNA in endothelial cells suggesting that the mechanism involves the regulation of ADAM17 protein expression leading to a decrease in ADAM17 activity and subsequently a decrease in sMCAM released. The decrease in MCAM RNA expression is probably related to the autoregulation of MCAM RNA expression induced by cleavage of sMCAM at the cell surface [45]. Thus, we conclude that soluble MCAM release can be rapidly decreased by inhibiting ADAM17 activities and, in the long term, NOX1 activities (Figure 8). Investigating the NOX1/ADAM17/MCAM interaction by using protein pull down and mass spectrometry, if NOX1 pharmacological inhibition modifies the protein complex or at least the NOX1 multiprotein enzymatic complex, the proteomic analysis of the cells supernatant to find other molecules downregulated by ADAM17 shedding activity inhibition may constitute a limitation of the study. It could highlight a broader mechanism not only involving MCAM but also other surface proteins shed by ADAM17 and involved in tumor development and angiogenesis (e.g., ICAM-1, VCAM-1, and others) [63].

From a clinical perspective, we found that MCAM, ADAM17, and NOX1 mRNA expressions are increased in the angiogenic, aggressive CMS4 subtype of CRC and that MCAM expression correlates with pro-angiogenic and pro-lymphangiogenic factors. Importantly, we demonstrated for the first time that sMCAM displayed pro-lymphangiogenic activities corroborating human findings.

## 5. Conclusions

Taken together, the main finding reported here is a new molecular mechanism involving NOX1 and ADAM17 in CRC development through the regulation of membrane MCAM expression and the proteolytic generation of sMCAM. These results highlight NOX1, ADAM17, and sMCAM as potential therapeutic targets in CRC. The elevated expression of MCAM in the angiogenic CMS4 CRC subtype raises the possibility of its inhibition for treating particularly aggressive CRC subtypes.

## Figures and Tables

**Figure 1 biomedicines-11-03185-f001:**
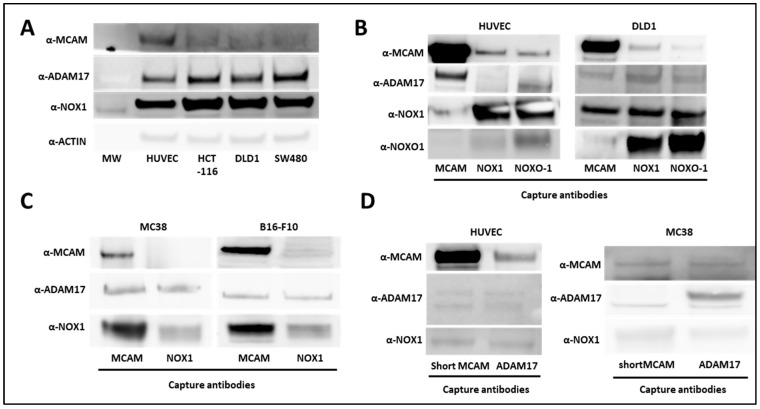
Association between NOX1, ADAM17, and MCAM on endothelial and cancer cells. (**A**) NOX1, ADAM17, MCAM, and actin protein expression in human vascular endothelial cells (HUVEC) and DLD1, SW480, and HCT-116 human colorectal cancer cell lines. The protein extracts of cell lines were analyzed by Western blot (*n* = 3). (**B**) MCAM, NOXO1, and NOX1 proteins were immunoprecipitated from HUVEC (left), and DLD1 (right) cell lysates with capture antibodies. Protein analysis of co-precipitated partners was analyzed by Western blot. MCAM, ADAM17, NOX1, and NOXO1 protein expressions were revealed with specific antibodies (*n* = 3). (**C**) MCAM and NOX1 proteins were immunoprecipitated from MC38 (left) and B16-F10 (right) cancer cell lysates with corresponding antibodies. Protein analysis of co-precipitated partners was analyzed by Western blot. MCAM, ADAM17, and NOX1 protein expression were revealed with specific antibodies (*n* = 3). (**D**) The short MCAM isoform and ADAM17 proteins were immunoprecipitated from HUVEC (left) and MC38 (right) cell lysates. Protein analysis of co-precipitated partners was performed by Western blot. MCAM, ADAM17, and NOX1 protein expressions were revealed with specific antibodies (*n* = 3).

**Figure 2 biomedicines-11-03185-f002:**
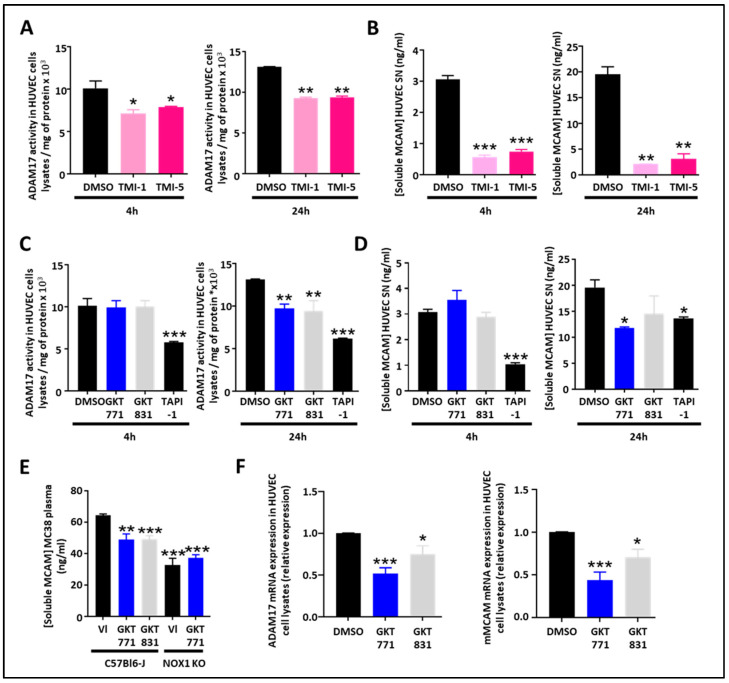
Pharmacological targeting of the NOX1/ADAM17 protein complex inhibits cleavage of MCAM, impairs ADAM17/MCAM expression and release of soluble MCAM, in HUVEC. (**A**,**B**) ADAM17 activity measured on HUVEC cell lysates and soluble MCAM content in HUVEC supernatant after treatment with ADAM17 inhibitors (TMI-1 and TMI-5, 10 µM). Measurement of ADAM17 activity in HUVEC cell lysates after short-term (4 h) or long-term (24 h) treatments are shown (**A**) (*n* = 5). Quantification of soluble MCAM by ELISA after short-term (4 h) or long-term (24 h) treatments are shown (**B**) (*n* = 5). (**C**,**D**) ADAM17 activity measured on HUVEC cell lysates and soluble MCAM content in HUVEC supernatant after treatment with NOX1 (GKT771 10 µM), NOX4 (GKT831 10 µM), and metalloprotease inhibitors (TAPI-1 10 µM). Measurements of ADAM17 activity in HUVEC cell lysates after short-term (4 h) or long-term (24 h) treatments are shown (**C**) (*n* = 5). Quantifications of soluble MCAM by ELISA after short-term (4 h) or long-term (24 h) treatments are shown (**D**) (*n* = 5). (**E**) Soluble MCAM content in mice plasma of WT and NOX1-deficient mice bearing MC38 tumors treated with NOX1 (GKT771) or NOX4 (GKT831) was analyzed by soluble MCAM ELISA. (**F**) Relative mRNA expression of ADAM17 (left) and MCAM (right) on HUVEC cell lysates after treatment with DMSO (control), NOX1 (GKT771 10 µM), and NOX4 (GKT831 10 µM) pharmacological inhibitors. * *p* < 0.05; ** *p* < 0.01; *** *p* < 0.001.

**Figure 3 biomedicines-11-03185-f003:**
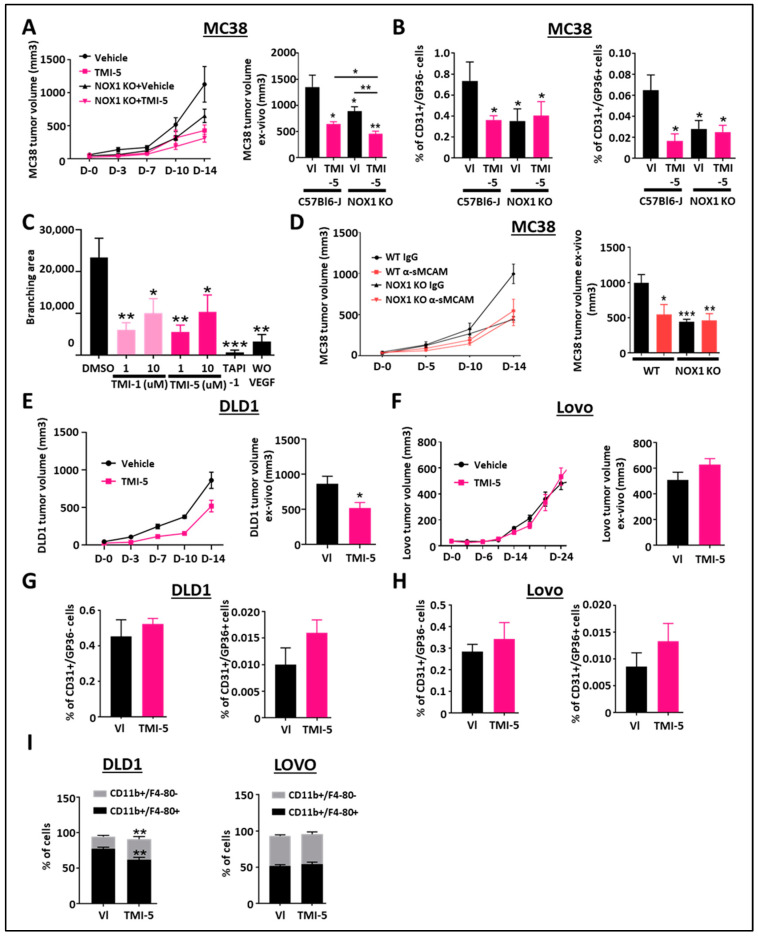
Pharmacological inhibition of ADAM17 reduces CRC tumor growth. (**A**) WT- and NOX1-deficient mice with established MC38 tumors were treated with TMI-5 (ADAM17 inhibitor, *n* = 8) or vehicle (VI) (*n* = 8). Tumor size was measured in vivo using a caliper on days 0, 3, 7, 10, and 14 (**A**, left). Tumor size was measured ex vivo at the end of the experiment on day 14 (**A**, right). (**B**) Tumor-associated blood vascular endothelial cells (CD45-/CD31+/GP38-) (left) and lymphatic endothelial cells (CD45-/CD31+/GP38+) (right) were analyzed by flow cytometry for the corresponding treated groups. (**C**) Aortic rings from C57/BL6 mice were stimulated with mouse rVEGF-A in the presence of TMI-1 (1 µM or 10 µM), TMI-5 (1 µM or 10 µM) or TAPI-1 (10 µM) and compared with the vehicle DMSO-treated group as the negative control (*n* = 7). The graphic shows the results of the quantification of the branching area. (**D**) WT and NOX1-deficient mice with established MC38 tumors were treated with anti-soluble MCAM antibody (*n* = 12) or corresponding control antibody (*n* = 12). Tumor size was measured in vivo using a caliper on days 0, 5, 10, and 14 (left). Tumor size was measured ex vivo at the end of the experiment on day 14 (right). (**E**,**F**) Immunodeficient NSG mice with established DLD1 or Lovo tumors were treated with TMI-5 (*n* = 9 for DLD1 and 8 for Lovo) or vehicle (VI) (*n* = 9). Tumor size was measured in vivo using a caliper on days 0, 3, 7, 10, and 14 for DLD1 experiments (**E**, left) and until day 24 for Lovo (**F**, left). Tumor size was measured ex vivo at the end of the experiment on day 14 for DLD1 (**E**, right) and day 24 for Lovo (**F**, right). (**G**,**H**) Tumor-associated blood vascular endothelial cells (CD45-/CD31+/GP38-) and lymphatic endothelial cells (CD45-/CD31+/GP38+) of DLD1 (**G**) and Lovo (**H**) tumors were analyzed by flow cytometry for the corresponding treated groups. (**I**) Pan-tumoral macrophages (CD45+/CD11b+/F4/80+) in DLD1 (left) and Lovo (right) tumors were analyzed by flow cytometry for the corresponding treated groups. * *p* < 0.05; ** *p* < 0.01; *** *p* < 0.001.

**Figure 4 biomedicines-11-03185-f004:**
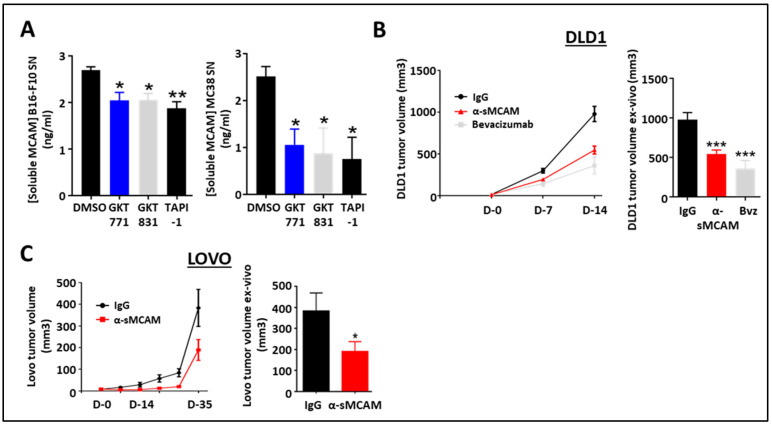
Blocking of sMCAM with specific monoclonal mAbs reduces human CRC tumor growth. (**A**) Soluble MCAM content in B16-F10 (left) and MC38 (right) cancer cell supernatants treated with NOX1 (GKT771, 10 µM), NOX1/4 (GKT831, 10 µM), and metalloprotease (TAPI-1, 10 µM) inhibitors were analyzed by soluble MCAM ELISA. (**B**,**C**) Immunodeficient NSG mice with established DLD1 (**B**) or Lovo (**C**) tumors were treated with anti-soluble MCAM antibody (*n* = 12), anti-human VEGF antibody bevacizumab (*n* = 7) or corresponding control antibody (*n* = 12) for DLD1 tumors and with anti-soluble MCAM antibody (*n* = 8) or corresponding control antibody (*n* = 9) for Lovo. Tumor size was measured in vivo using a caliper at days 0, 7, and 14 for DLD1 experiments (**B**, left) and until day 35 for Lovo (**C**, left). Tumor size was measured ex vivo at the end of the experiment on day 14 for DLD1 (**B**, right) and day 35 for Lovo (**C**, right). * *p* < 0.05; ** *p* < 0.01; *** *p* < 0.001.

**Figure 5 biomedicines-11-03185-f005:**
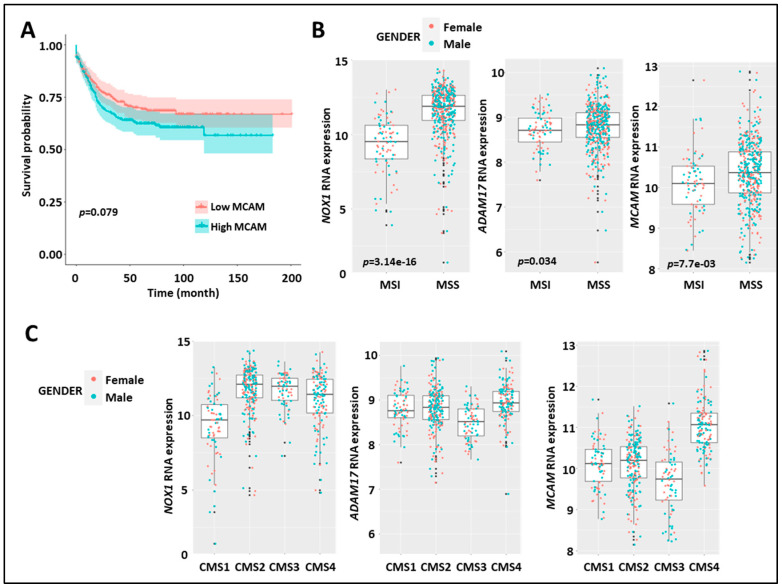
Expression of MCAM in human CRC correlates with shorter relapse-free survival, MMS status, and CMS4 subtype. (**A**) Kaplan–Meier survival analysis of the index of MCAM mRNA expression (high (red) vs. low (blue)) relative to the median value in CRC-patient tumor samples. (**B**) Expression of NOX1 (left), ADAM17 (middle), and MCAM (right) mRNA in CRC-patient tumor samples according to microsatellite instable/microsatellite stable status (MSI/MSS). (**C**) Expression of NOX1 (left), ADAM17 (middle), and MCAM (right) in the four consensus molecular subtypes (CMS 1–4) of human CRC.

**Figure 6 biomedicines-11-03185-f006:**
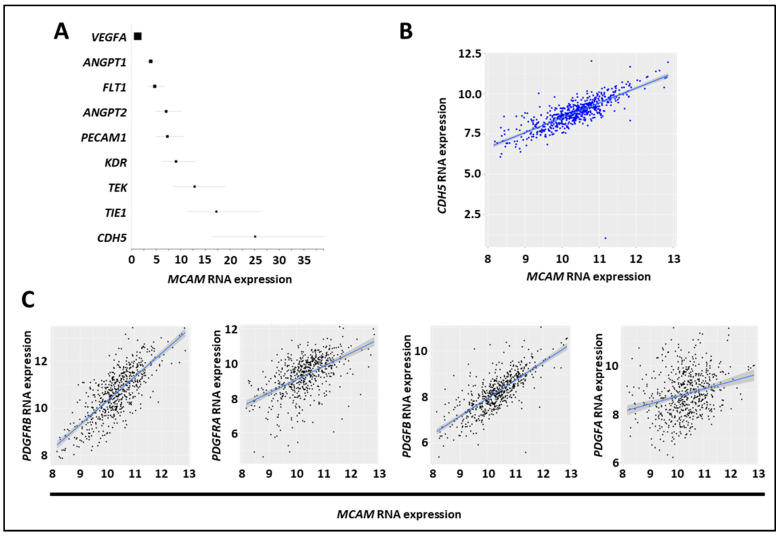
MCAM expression in CRC correlates with expression of angiogenic factors. (**A**) Forest plot showing the results of a meta-analysis of the association between MCAM mRNA expression and that of tumor-associated angiogenic molecules in CRC tumor samples. All markers, except VEGF-A, exhibited a positive association (according to the odds ratio, OR) and a positive correlation with MCAM mRNA expression. (**B**) Scatter plots with linear regression lines (blue) show the correlation between levels of MCAM mRNA and CDH5 in CRC samples. Levels of MCAM and CDH5 mRNA in different subgroups of patients were compared using a *t*-test with equal variance. The correlation between the expression of MCAM and CDH5 was calculated using Pearson’s and Spearman’s rank correlation analyses. (**C**) Scatter plots with linear regression lines (blue) show the correlation between levels of MCAM mRNA and PDGF receptor β, PDGF receptor α, PDGF-B, and PDGF-A in CRC samples. Levels of MCAM and the four different mRNA in different subgroups of patients were compared using a *t*-test with equal variance. The correlation between the expression of MCAM and the four different mRNA was calculated using Pearson’s and Spearman’s rank correlation analyses.

**Figure 7 biomedicines-11-03185-f007:**
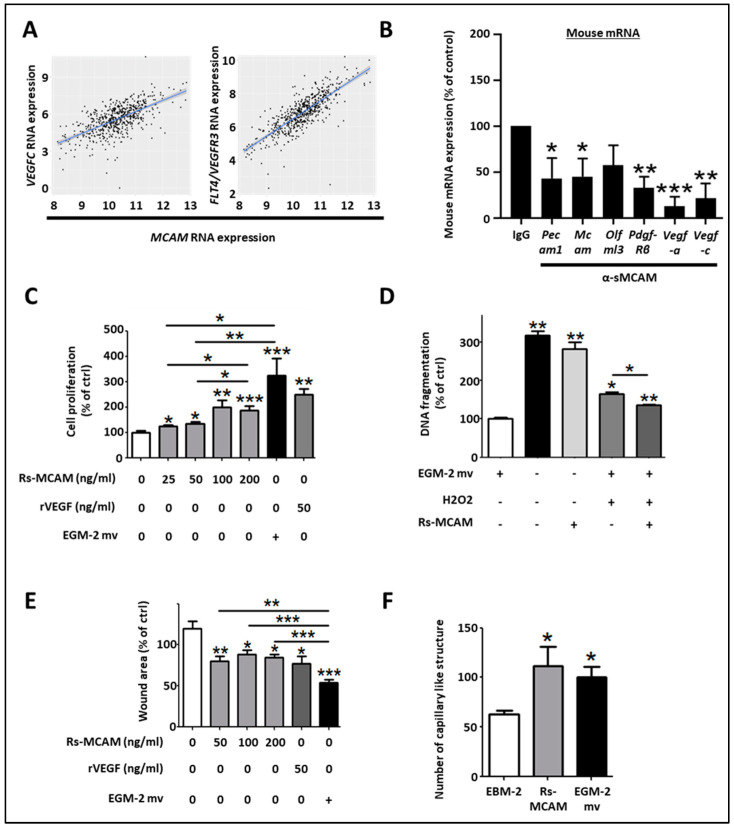
Soluble MCAM increases the lymphangiogenic activity of human dermal lymphatic endothelial cells. (**A**) Scatter plots with linear regression lines (blue) showing the correlation between levels of MCAM mRNA and VEGF-C, and FLT4 in CRC samples. Levels of MCAM and the two different mRNAs in different subgroups of patients were compared using a *t*-test with equal variance. The correlation between the expression of MCAM and the two different mRNAs was calculated using Pearson’s and Spearman’s rank correlation analyses. (**B**) Specific mouse mRNA expression in tumors was explored by NanoString arrays using specific mouse probes. Mouse mRNA expression of tumors treated with sMCAM mAb was calculated and compared to control mAb-treated mice as a control. Results are expressed as a percentage of control. (**C**–**F**) The in vitro pro-lymphangiogenic properties of soluble MCAM (sMCAM) were monitored by cell proliferation, apoptosis, wound healing, and 2-D capillary-like structure assays. Analysis of cell proliferation by BrdU incorporation in HDLEC was monitored after dose-response stimulation with rs-MCAM (25 to 200 ng/mL), recombinant VEGF (rVEGF), and in EGM2-mv complete culture medium (**C**) (*n* = 5). Results are expressed as a percentage of control. Analysis of rs-MCAM anti-apoptotic activity on HDLEC under chemically induced apoptosis by H_2_O_2_ was performed by DNA fragmentation assays (**D**) (*n* = 4). Results are expressed as a percentage of control. HDLEC motility was assessed by wound healing experiments (**E**). Cells were exposed to dose–response stimulation of rs-MCAM (50 to 200 ng/mL), rVEGF (50 ng/mL), or EGM2-mv complete culture medium. The percentage of wound closure was calculated based on the wound area at T = 0 h and T = 6 h using Image J software (*n* = 6). Results are expressed as a percentage of control. HDLEC’s ability to form 2-dimensional capillaries on coated Matrigel was assessed under stimulation with recombinant soluble MCAM (100 ng/mL) or complete EGM2-MV medium (**F**). The number of capillary-like structures was analyzed using Image J software (*n* = 5). * *p* < 0.05; ** *p* < 0.01; *** *p* < 0.001.

**Figure 8 biomedicines-11-03185-f008:**
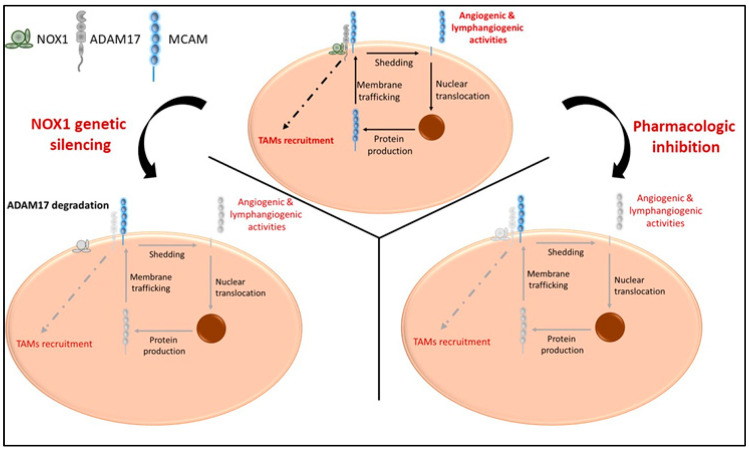
Proposed model illustrating the cellular effects of NOX1 and ADAM17 targeting ADAM17, soluble MCAM generation and pro-tumorigenic functions. In naïve tumors, NOX1 and ADAM17 are associated and induce release of soluble MCAM with pro-angiogenic ability. The release of soluble MCAM induces cellular events leading to the production of growth factors including MCAM itself. In addition, ADAM17 mediates events involved in the recruitment of TAMs (**upper panel**). NOX1 genetic silencing (**lower left panel**) impaired ADAM17 expression and activity resulting in a decrease in soluble MCAM release and TAM recruitment. NOX1 and ADAM17 pharmacologic inhibition (**lower right panel**) impaired shedding of MCAM, release of soluble MCAM, production of MCAM molecules, and TAM recruitment. Collectively, these events impaired tumor angiogenesis, TAM recruitment, and CRC tumor growth.

## Data Availability

Data are available upon request from the corresponding author(s).

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
