# Peer review of "Targeting of the NOX1/ADAM17 Enzymatic Complex Regulates Soluble MCAM-Dependent Pro-Tumorigenic Activity in Colorectal Cancer"

_biomedicines, 2023, doi:10.3390/biomedicines11123185_

Round 1
Reviewer 1 Report
Comments and Suggestions for Authors
The researchers determined the participation of NOX1 and ADAM17 in the generation of sMCAM from cancer cells and blood vascular endothelial cells. Their findings showed the contribution of NOX1 and ADAM17 in tumor angiogenesis and which is related to cancer progression. An important contribution is the clinicopathological relevance of the expression of NOX1, ADAM17 and MCAM in colorectal cancer.
The title of the manuscript is appropriate, clear and has an adequate number of words. The summary is well formulated, "the contextualization of the topic could be improved by a couple of initial lines", however, the objective, the justification are well developed, a brief development of the methodology and the most relevant results is presented, ending with a short conclusion.
The introduction is well written, the contextualization of the topic is well founded, the knowledge gap is well defined and the objective of the research is well described.
Regarding the methodology, the procedures are very well described, which guarantees the quality of the results, in addition to allowing the experiments to be replicated in other laboratories.
As for the results and their discussion, they are conclusive, the results are well described and the discussion is well founded. Having said this, the figures are not up to the quality of the manuscript, I recommend improving the resolution of all the images, some of them are illegible.
The conclusions are short, but they are forceful, and are related to the objective of the research.
Page 22. remove the MDPI logo
I did not observe the report of an ethics committee, if they have it, the approval of the project by an ethics committee must be reported.
Reviewer 2 Report
Comments and Suggestions for Authors
Targeting of the NOX1/ADAM17 enzymatic complex regulates soluble MCAM-dependent protumorigenic activity in colorectal cancer, extended findings on the possible physical association of NOX1 with ADAM17, stabilizing ADAM17 and promoting proteolytic processing of MCAM. This seems to lead to tumor growth via angiogenesis and other pathways.
The second half of the paper, showing effects of ADAM17 and cleaved MCAM is reasonably convincing.
The section on NOX1, itself, is problematic.
One major problem with the paper is the fragmented Materials and Methods writing. Much of what should be contained in the M&M section is scattered across the text and in the figure legends. Cell lines just show up in the text and figure without mention in the M&M section and without mention of sources of the lines. If they were used in prior work, please just list the cell lines in the M&M section with a reference. If you insist on this scattered presentation, mention this in the M&M section so the reader is prepared. Also, there is no mention of drug treatments in the M&M section. Also, please define HDLECS.
In the results section, 3.1, list the reference for your prior observations for NOX1, ADAM17 and mMCAM (25?) there.
I find the data on the NOX1 and ADAM17 in Fig 1 less than completely convincing. I have seen such experiments where the investigators used mass-spec methods to find the full list of pulled down proteins and often see a huge number of such proteins. Sometimes the determination of relevant associations from such lists seems arbitrary, based more on the researchers’ goals rather than being objective. A negative and/or positive control is needed in this study. I was hoping that by looking at the reference for the prior suggestion of a physical association, I would find a more convincing demonstration. However, the relevant panels in fig. 6 of that paper are cut off. However, the main point found in panel b is there-low NOX1 results in destabilization of ADAM17. However, this could be mediated by superoxide or hydrogen peroxide generation in the vicinity of NOX1. In regard to this, how does GKT771 inhibit NOX1? I suspect that it might impact assembly of the holoenzyme. Does this disrupt the proposed physical interaction? Does the inhibitor impact the pull-down results, and should this be expected?
The main impact of the GKT771 seems to be via the mRNA levels of mMCAM. In panel 2F, what was the time span of drug treatment and in panel E, what are the mRNA of MCAM in the tumors in each of the conditions?
Comments on the Quality of English Languageok
Round 2
Reviewer 2 Report
Comments and Suggestions for Authors
Somewhat improved. OK to publish.